# Classification Prediction of Alzheimer’s Disease and Vascular Dementia Using Physiological Data and ECD SPECT Images

**DOI:** 10.3390/diagnostics14040365

**Published:** 2024-02-07

**Authors:** Yu-Ching Ni, Zhi-Kun Lin, Chen-Han Cheng, Ming-Chyi Pai, Pai-Yi Chiu, Chiung-Chih Chang, Ya-Ting Chang, Guang-Uei Hung, Kun-Ju Lin, Ing-Tsung Hsiao, Chia-Yu Lin, Hui-Chieh Yang

**Affiliations:** 1Department of Radiation Protection, National Atomic Research Institute, Taoyuan 325, Taiwan; 2Division of Behavioral Neurology, Department of Neurology, National Cheng Kung University Hospital, College of Medicine, National Cheng Kung University, Tainan 701, Taiwan; 3Institute of Gerontology, National Cheng Kung University, Tainan 701, Taiwan; 4Alzheimer’s Disease Research Center, National Cheng Kung University Hospital, Tainan 704, Taiwan; 5Department of Neurology, Show Chwan Memorial Hospital, Changhua 500, Taiwan; 6Department of Neurology, Institute of Translational Research in Biomedicine, Kaohsiung Chang Gung Memorial Hospital, Chang Gung University College of Medicine, Kaohsiung 833, Taiwan; 7Department of Nuclear Medicine, Chang Bing Show Chwan Memorial Hospital, Changhua 505, Taiwan; 8Healthy Aging Research Center and Department of Medical Imaging and Radiological Sciences, College of Medicine, Chang Gung University, Taoyuan 333, Taiwan; 9Molecular Imaging Center and Department of Nuclear Medicine, Chang Gung University and Chang Gung Memorial Hospital, Taoyuan 333, Taiwan

**Keywords:** ECD SPECT images, Alzheimer’s disease, vascular dementia, classification prediction

## Abstract

Alzheimer’s disease (AD) and vascular dementia (VaD) are the two most common forms of dementia. However, their neuropsychological and pathological features often overlap, making it difficult to distinguish between AD and VaD. In addition to clinical consultation and laboratory examinations, clinical dementia diagnosis in Taiwan will also include Tc-99m-ECD SPECT imaging examination. Through machine learning and deep learning technology, we explored the feasibility of using the above clinical practice data to distinguish AD and VaD. We used the physiological data (33 features) and Tc-99m-ECD SPECT images of 112 AD patients and 85 VaD patients in the Taiwanese Nuclear Medicine Brain Image Database to train the classification model. The results, after filtering by the number of SVM RFE 5-fold features, show that the average accuracy of physiological data in distinguishing AD/VaD is 81.22% and the AUC is 0.836; the average accuracy of training images using the Inception V3 model is 85% and the AUC is 0.95. Finally, Grad-CAM heatmap was used to visualize the areas of concern of the model and compared with the SPM analysis method to further understand the differences. This research method can quickly use machine learning and deep learning models to automatically extract image features based on a small amount of general clinical data to objectively distinguish AD and VaD.

## 1. Introduction

In populations aged 65 and older, Alzheimer’s disease (AD) stands out as the most prevalent type of dementia. Vascular dementia (VaD) ranks as the second most common type [1]. From therapeutic and prognostic perspectives, distinguishing between AD and Subcortical Ischemic Vascular Dementia (SIVD) is important. Notably, the incidence and mortality rates of VaD, inclusive of SIVD, are significantly higher compared to pure AD, primarily due to its more frequent association with vascular risk factors and mobility disorders. Further, VaD is associated with a 50% reduction in median survival (from 6–7 years down to 3–4 years), increased medical costs, higher rates of comorbidity, institutionalization, and the need for caregiving services [2].

Demographic studies indicate that, among the elderly population, the probability of females developing AD is slightly higher than that of males. This elevation is primarily attributed to an increased risk of AD after age-adjustment, with a relative risk (RR) of 1.3 [3]. Conversely, VaD, stroke, and other atherosclerotic cardiovascular diseases are more prevalent in males [3]. It is mentioned in the literature [4] that VaD represents a heterogeneous group of dementias caused by ischemic, hemorrhagic, anoxic, and hypoxic brain damage. Ischemic VaD may arise from macrovascular or microvascular cerebral disease, or a combination of both, while hemorrhagic VaD is often associated with hypertension, cerebral amyloid angiopathy, and intralobar hemorrhages. On the other hand, AD involves neurofibrillary tangles, amyloid plaques, and neuronal death, classifying it as a neurodegenerative disease. This typically manifests as memory loss and cognitive decline, with the condition progressively worsening over time.

Several studies have indicated that the co-occurrence of AD and cerebrovascular pathologies is referred to as mixed dementia [5]. Each pathology contributes to different degrees of impact, leading to various stages of the disease in patients, with pure AD and pure VaD representing the opposite ends of the spectrum [6,7]. In addition to the possible common vascular etiopathogeny in AD and VaD, the Nun study demonstrated that amyloid/tau proteins and vascular burden are neuropathologically related in the autopsied brains of most patients with dementia [8]. Because these neuropathological and epidemiological studies indicate that the pathologies of AD and VaD are so associated, mixed dementia is considered the most prevalent subtype.

Some studies have used MRI imaging to differentiate between AD and VaD. Typically, cortical atrophy in AD primarily affects the temporal and parietal lobes, with mesial temporal regions and the precuneus/posterior cingulate cortex being especially impacted. In the early stages of the disease, there is a degeneration of white matter (WM) bundles [9]. On the other hand, the regional pattern of cerebrovascular lesions in VaD varies across its subtypes [10,11]. Most commonly in small vessel disease, which leads to subcortical VaD, multifocal lesions are observed on qualitative MRI. These are predominantly located in the deep gray matter and primarily in the frontal periventricular and subcortical WM [12,13]. Efforts to distinguish AD from VaD using diffusion MRI [14,15] have yielded inconsistent findings. Additionally, Arterial Spin Labeling Magnetic Resonance Imaging (ASL-MRI) employs arterial blood as an endogenous contrast for detecting regional cerebral blood flow (CBF) perfusion in patients. In comparing AD and VaD, one study has demonstrated differential patterns of CBF reduction in the frontal and parietal cortices [16], while another study has indicated reductions in the frontal and temporal regions [17]. Furthermore, recent years have seen a growing number of studies leveraging biological data, such as biomarkers and marker genes, to delve into the distinctions between AD and VaD [18,19,20]. This approach aids in enhancing our comprehension of disease pathways and mechanisms, and serves to validate prevailing hypotheses.

In the clinical diagnosis of AD and VaD in Taiwan, in addition to the demographic data and physiological data obtained from clinical inquiries and laboratory tests, the Tc-99m-ECD SPECT imaging examination is included in the dementia imaging examination items covered by Taiwan’s national health insurance. Brain perfusion SPECT images should be helpful for the diagnosis of AD and VaD, but relevant studies that used nuclear medicine images to distinguish AD and VaD could not be found. If the information from the clinical routine examination can be used by machine learning and deep learning technologies that are widely used at present, and high-dimensional analysis and calculation of features can be introduced to increase their ability to distinguish types of dementia, it will be of direct benefit to clinical practice. In view of this, the focus of this study is to investigate the use of demographic and physiological data or Tc-99m-ECD SPECT images to distinguish AD from VaD.

## 2. Materials and Methods

This study primarily investigated whether the use of physiological data and Tc-99m-ECD SPECT images can potentially distinguish between AD and VaD when trained using AI models. The physiological data is categorized as structured data, for which we employed the SVM (support vector machine) method to achieve our classification objectives. For the Tc-99m-ECD SPECT images, we utilized the CNN model for differentiation between AD and VaD. The entire data training process is shown in Figure 1.

### 2.1. Subjects

Data for this study were sourced from the Taiwanese Nuclear Medicine Brain Image Database, established by the Institute of Nuclear Energy Research (now renamed as the National Atomic Research Institute). This database, backed by government support, is a prospective multi-center clinical study led by M.-C.P that compiles local data from four medical centers. It holds significant representative value for dementia research within the Taiwanese population. This study utilized data from 112 AD patients and 85 VaD patients within the database. These data include 13 basic information items, 17 laboratory examinations, 3 scale examinations, Tc-99m-ECD SPECT imaging, and diagnostic results. All the aforementioned subjects underwent evaluations by neurologists and clinical psychologists, completed medical history inquiries (including crucial systemic and brain disease histories, and clinical dementia assessment scales), and were diagnosed by clinical neurologists. Their imaging data were interpreted by nuclear medicine physicians. The complete process of clinical data collection and its utilization received approval from the Institutional Review Board (IRB).

### 2.2. Statistical Analysis and Normalization of Physiological Data

Statistical analysis was performed on 33 sets of physiological data to understand the data distribution and differences between the AD and the VaD groups. For continuous variables, normality was tested using the Kolmogorov–Smirnov test. If the data followed a normal distribution, the One-way ANOVA was used to detect differences between populations. If the distribution was not normal, the Kruskal-Wallis test was used to determine differences. For categorical variables, the chi-squared test was used to assess differences between the groups. All of the above were implemented using the Scipy library in Python.

During the analysis process, we observed an imbalance in the number of males to females within the VaD population (57:28). Additionally, some characteristics exhibited varying numerical ranges due to gender, thereby affecting the analysis results. In order to solve the above problems, we segregated the continuous variable items based on gender into two groups: male and female. We then applied z-score normalization to each group, adjusted the distribution of male and female, and subsequently re-established the distribution of the AD and VaD groups. All subsequent analyses and disease classification training utilized the normalized feature values. Statistical analyses of the characteristic items with significant differences between the two disease groups are marked in bold. In addition, the box plot results of the six items that still had significant differences after normalization were compared before and after normalization.

### 2.3. Machine Learning for Physiological Data

We consider that the SVM method is quite stable among machine learning techniques and often achieves excellent accuracy in classification tasks. Therefore, we used this method to train our data, seeking to understand the contributions or influences of various physiological data in differentiating between the AD and VaD groups. We utilized the scikit-learn library in Python, employing a linear kernel and setting the regularization parameter to 1.0. Although the model was trained using the most basic parameter settings, the absence of highly nonlinear mapping means that it is easier to intuitively explain model training results using feature weights. In addition, due to the limited data available in this study, recursive feature elimination (RFE) combined with 5-fold cross-validation was also applied to select the most relevant features. By reducing the number of feature items used for training, we expected to improve the discriminative accuracy between the AD and VaD groups.

### 2.4. Image Acquisition and Processing

The Tc-99m-ECD SPECT images used in this study were sourced from four medical centers and obtained using E-CAM, Symbia T16, and Symbia T2 SPECT instruments (Siemens Medical Solutions, Malvern, PA, USA) with LEHR (low energy high resolution) and fan beam collimators. Although different imaging equipment was utilized, the image acquisition procedures are similar, and through spatial normalization and registration to the SPECT perfusion template, all images were pre-processed and resampled to 95 × 95 × 68 with the voxel size 2 × 2 × 2 mm^3^. We retained only slices located within the brain parenchyma, and these 3D images were organized into three groups of 2D images, each consisting of 4 × 4 slices. The size of each 2D image was 580 × 580. For detailed processing methods and schematic diagrams, please refer to Chapter 2.2 of the literature [21].

### 2.5. Deep Learning for ECD Image

The Inception V3 model was chosen to perform the classification task of distinguishing between the AD and VaD groups using Tc-99m-ECD SPECT images. A fully connected layer (FC) with a length of 128 was attached to the top layer of the Inception V3 model. Subsequently, a batch normalization (BN) layer and a dropout layer were added, with the dropout rate set at 0.5. For the detailed model architecture, please refer to Figure 1 of literature [22]. The categorical cross-entropy was utilized as the loss function, while the Adaptive Moment Estimation (Adam) [23] served as the optimization algorithm. The learning rate and batch size were set at 0.00001 and 32, respectively, for model training. An early stopping mechanism was employed to determine the appropriate stopping point and to select the optimal epoch. Twenty Tc-99m-ECD SPECT images (AD = 9; VaD = 11) were randomly chosen as an independent test set. Out of the remaining 177 images (AD = 103; VaD = 74), 80% were allocated for training and 20% for validation. For data augmentation, the ranges for random width and height shifts were set between ± 10%, and the zoom range was adjusted between ±8%. The development environment was set up using Python 3.7, with Keras 2.4 to build neural networks and import pre-trained models. The backend operated on TensorFlow 1.15.2 (Google, Mountain View, CA, USA).

### 2.6. Model Training and Evaluation

Whether using physiological data or Tc-99m-ECD SPECT images, the performance evaluation indicators for distinguishing AD and VaD through model training are the same. The accuracy of the model was evaluated using the receiver operating characteristic (ROC) curves and the area under the curve (AUC). The ROC curve, with 95% confidence intervals (CI), was plotted using MATLAB (MATLAB R2020a, MathWorks, Natick, MA, USA) and was derived from 1000 bootstrap iterations. Furthermore, statistical analysis was conducted on the classification prediction results. This analysis included the calculation of sensitivity, specificity, precision, accuracy, and the F1 score. For the above evaluations, VaD was defined as positive.

### 2.7. Interpreting Models with Grad-CAM

By employing Grad-CAM with Inception V3 model, the gradients are being propagated to the last convolutional layer, and to weight the forward activation maps. The class-discriminative localization map, obtained through weighted combinations, localizes relevant image regions and reveals the influences on the class of interest as a heatmap. Since the input data is pre-processed with the grid method, it is necessary to apply an inverse-preprocessing step to restore the heatmaps to their original dimensions. Subsequently, registration with a brain atlas was carried out, and voxel values exceeding the 90% maximum threshold were counted with respect to various brain regions and two different groups. Finally, we compared and attempted to establish connections between these explanatory results and insights from neurologists.

Additionally, we also analyzed the Tc-99m-ECD SPECT images of both AD and VaD groups using the SPM12 (Statistical Parametric Mapping) tool. A two-sample *t*-test statistical method was utilized to compare the significantly different regions between the two groups. We set the *p*-value at 0.01 (*p* < 0.01), and the cluster size at 400 (cluster size > 400), and considered the effects caused by age as covariates, subsequently excluding them. The analysis ultimately highlighted brain regions where each group significantly differed from the other. These findings serve as a reference when compared with the results from deep learning.

## 3. Results

### 3.1. Characteristics of Demographic Data and Physiological Data

Statistical analysis was performed on 33 demographic and physiological data items for AD and VaD patients from the Taiwanese Nuclear Medicine Brain Image Database. The results are presented in Table 1. In the original data, values indicating significant differences between the AD and VaD groups (*p* < 0.05) are marked in bold. In addition, in response to the unbalanced ratio of male to female in the VaD population (57:28) and the fact that some features have different numerical ranges due to gender, the features of continuous variables were normalized to improve the above-mentioned effects. Post-normalization, values indicating significant differences (*p* < 0.05) between the AD and VaD groups are also marked in bold.

The data in the feature columns maintained significant differences between the AD/VaD groups both before and after normalization, including age, height, body weight, WBC (white blood cell), HDL (high-density lipoprotein), creatinine, and folic acid. Two features, height and body weight, revealed obvious differences between the AD/VaD groups, largely due to the differing male-to-female ratios in each group. After z-score normalization based on the gender distribution, these features still presented significant differences. However, the magnitude of the disparity between the two groups for these features was substantially reduced. Figure 2 presents six significantly different features among the continuous variables and provides a comparison of their distributions before and after normalization using a boxplot.

### 3.2. Using Machine Learning to Classify AD/VaD Using Physiological Data

We used the SVM method to train the possibility of distinguishing AD and VaD through physiological data. In addition, RFE combined with 5-fold cross-validation was also used to select the optimal number of features. Using accuracy as the metric for correct classification, the highest accuracy rate of 76.2% was achieved when the top 19 most correlated features were considered. Table 2 lists the training results of both the SVM model with all 33 features and the SVM model with the 19 most relevant features, including the results of 5-fold and its average training performance indicators. Figure 3 shows the results of the ROC curves and the AUC.

### 3.3. Using Deep Learning to Classify AD/VaD Using Tc-99m-ECD SPECT Images

Besides using physiological data to train the model to distinguish between AD and VaD, we also used Tc-99m-ECD SPECT images to train the deep learning model Inception V3 to distinguish AD and VaD. We evaluated the performance of model training on a randomly selected independent test set (20 images) and presented the results of 5-fold and average training performance indicators in Table 3. In the results of ensemble learning using the averaging approach, the sensitivity, specificity, precision, accuracy, and F1 score were 81.82%, 88.89%, 90%, 85%, and 85.71%, respectively. The ROC curves of the above training results are presented in Figure 4, the average AUC value reaches 0.95.

### 3.4. Correlation between Model Interpretation and Brain Regions

The Inception V3 model’s differentiation between AD and VaD was visually inspected using the Grad-CAM map to identify regions deemed critical by the model for classification decisions. For all test data, a Grad-CAM map was calculated for each patient. However, to provide a comprehensive overview of the regional differences in the class-feature heatmap between AD and VaD groups, the averaged results were used to present the Grad-CAM map for the two diseases (as shown in Figure 5). From the results, it is evident that the AD group focuses more on the lower slices of the brain, while the VaD group tends towards the upper slices. In terms of brain lobes, it can be generally observed that AD leans towards the occipital lobe region, while VaD favors the temporal lobe region. Figure 6 displays detailed pixel count statistics mapped to 91 brain regions. In Table 4, we have listed the top 10 brain regions from the aforementioned results and compared them with the top 10 brain regions analyzed using the SPM statistical method. For the AD group, both the trained model’s Grad-CAM map and the traditional analytical method highlighted the precuneus and cuneus regions. In contrast, for the VaD group, there was no clear corresponding brain region.

## 4. Discussion

In this study, we initially utilized demographic data and physiological data to distinguish between the AD and VaD groups. We began with statistical methods for analysis and then adopted machine learning techniques to further investigate its capability in disease classification. Given that demographic and physiological data are the most accessible information for patients, if promising features that differentiate these two diseases can be identified from this data, then when clinicians face ambiguous or uncertain scenarios during the diagnostic process, they can refer to these features, thereby making their diagnoses with greater confidence.

In Figure 2, we present boxplots comparing six features before and after normalization. Regardless of whether it was before or after normalization, five of these features exhibited significant differences between the AD and VaD groups. Notably, the feature free T4 showed no distinct differentiation between the AD and VaD groups before normalization (*p* = 0.065); however, after normalization, the difference became significant (*p* = 0.036). We also display the boxplots of this feature before and after normalization. Examining these six features more closely, the age feature, whether before or after normalization, revealed a pronounced median age difference exceeding five years between the AD and VaD groups. Even though the data range and dispersion for both groups were very similar, the data distribution for the AD group leaned towards higher values. This can be attributed to the fact that AD, being a neurodegenerative disease, is more commonly observed in an older age group [24]. On the other hand, VaD is associated with external factors like occlusions or bleeding in the cerebral vessels, which might explain the slightly younger average age of its patients. For the WBC feature, there is a noticeable difference in the medians of the AD and VaD groups, both before and after normalization, even though their data distribution ranges are similar. Previous research indicates a correlation between the WBC count and cardiovascular diseases [25,26]. However, we need more information and in-depth analysis to understand the reasons for this result. Regarding the HDL feature, before and after normalization, it is observed that the median and distribution for the VaD group are both lower than those for the AD group. It is widely recognized that lower levels of HDL can increase the risk of atherosclerosis, which significantly impacts cardiovascular health [27]. This observation aligns with the slightly lower data noted for the VaD group. However, in this database, the distribution of HDL values for the AD group is broader. Further examination and investigation of individual cases are needed to gain a deeper understanding. For the creatinine feature, before and after processing, we observed that the median and distribution for the VaD group were higher than those of the AD group. Previous studies have indicated that patients with higher levels of creatinine face an increased risk of heart disease and stroke [27]. This observation reasonably explains the slightly higher data in the VaD group. Since the normal reference range for creatinine testing varies between males and females, our study considered the different gender ratios between the AD and VaD groups and applied normalization to mitigate this influence. In the case of the free T4 feature, the AD group’s median value was higher, with a more concentrated data distribution. Some previous studies have mentioned that elevated free T4 levels, even within the normal range, might be associated with an increased risk of heart disease, and also potentially cognitive decline [28]. However, in this study, given the absence of more detailed pertinent information for reference, we can only depict these differences based on the available data. Lastly, regarding folic acid, we observed that both the median and the overall distribution were slightly higher in the AD group compared to the VaD group. Past research has shown that folic acid can aid in reducing homocysteine in the blood [29]. Higher homocysteine levels, considered a risk factor for cardiovascular disease, are also associated with cognitive impairment [30]. Therefore, we cannot definitively account for the observed differences in folic acid levels between the AD and VaD groups, and we present our findings based on the data at hand.

Further to statistically exploring the features in the demographic data and physio-logical data that significantly differ between the AD and VaD groups, these features were further processed using machine learning for classification and prediction. Initially, we trained using the complete set of 33 features and utilized 5-fold cross-validation to evaluate the model’s stability and reliability. The results in Table 2 indicate that the average sensitivity, specificity, precision, accuracy, and F1 score were 50.59%, 83.93%, 70.49%, 69.54%, and 58.90%, respectively. This trial’s performance was suboptimal, especially in terms of sensitivity, pointing to its limited accuracy in detecting VaD. However, its capacity to identify AD was commendable. Given the limited data size in this study, we aimed to enhance prediction accuracy by selecting the optimal feature number using RFE. Subsequently, the most important 19 features were employed for SVM model training and validation. The 5-fold cross-validation average results for sensitivity, specificity, precision, accuracy, and F1 score were 71.76%, 88.39%, 82.43%, 81.22%, and 76.73%, respectively. Figure 3A,B illustrates the ROC curve and AUC values of 5-fold cross-validation for the SVM model, comparing the complete 33 features against the selected 19. The average AUC for the complete 33 features was 0.75 with a standard deviation of 0.1, while the 19 features had an average AUC of 0.836 with a standard deviation of 0.06. The results suggest that feature selection considerably enhanced the efficiency and stability of distinguishing between AD and VaD.

Another aspect of this study employs deep learning techniques to differentiate be-tween AD and VaD using Tc-99m-ECD SPECT images. The ability to distinguish between these diseases using images is notably superior to relying solely on demographic and physiological information. This superiority is evident in various performance evaluation metrics, especially the AUC value. Specifically, the AUC result when using images for classification prediction stands at 0.95, compared to 0.836 when using only physiological information.

While the results obtained using the Tc-99m-ECD SPECT images to train the Inception V3 model to distinguish AD from VaD are impressive, neurologists are keen to understand which brain features influence the model’s decisions and whether these align with expert opinions or current knowledge. In this regard, we employed the Grad-CAM heatmap to visualize the pixels that heavily influence the model’s decision-making process. As depicted in Figure 5, the heatmaps for the AD and VaD groups differ significantly. Unnatural square edges are visible at the boundaries between different slices in the image. This appearance is due to the representation of 48 slice images simultaneously, achieved by merging three sets of equally spaced 2D images. Given our focus on analyzing pixels within the brain, this peripheral information can be disregarded. To gain a deeper insight into the brain regions highlighted by the Grad-CAM heatmap, we registered the image with the standard AAL brain template and tallied the number of pixels that lit up in its 91 brain regions. We defined ‘lit-up’ areas as those pixels with values exceeding 90% of the maximum. It is worth noting that, in general, the Grad-CAM heatmap provides a broad representation of locations and might not be highly precise. Nevertheless, we endeavored to analyze and count the highlighted pixels within each brain region to discern any potentially meaningful insights.

Figure 6 and Table 4 display the histogram representing the count of lit-up areas, along with the top ten rankings of the average Grad-CAM heatmap in each brain region for both AD and VaD groups. The color coding in Table 4 is based on proportion: a white background indicates greater than two-thirds of the maximum value; light gray represents greater than one-third of the maximum value; and dark gray represents less than one-third of the maximum value. On the Grad-CAM heatmap, the occipital region of the AD group had the largest representation, followed by the cuneus and precuneus. In contrast, the temporal region of the VaD group had the most significant presence, followed by the lingual, cerebellum, and putamen.

Original Tc-99m-ECD SPECT images, when analyzed using SPM, reveal pixels with significant differences between AD and VaD in specific brain regions. The majority of the areas where AD > VaD are in the thalamus, adjacent to the hippocampus, as well as the cuneus and precuneus. Many regions where VaD > AD are found in the pre-central, accompanied by the nearby post-central gyrus, as well as the frontal region. The cerebral blood flow imaging pattern in VaD patients shows asymmetrical and variable hypoperfusion, while AD exhibits more standardized patterns, such as hypoperfusion in bilateral temporal and parietal regions. Previous research indicated that bilateral hypoperfusion was detected in the temporal and/or parietal regions in 33% of VaD patients and 70% of AD patients when using Tc-99m-HMPAO SPECT for both AD and VaD groups [31]. An analysis of autopsied human brain tissue from literature examining the differences between AD and VaD also mentioned employing the ratio of myelin-associated glycoprotein to proteolipid protein-1 (MAG:PLP1) and the vascular endothelial growth factor-A (VEGF) index to observe cerebral hypoperfusion patterns. In this analysis, the MAG:PLP1 ratio in the frontal and parietal cortex of the VaD group was the lowest (indicative of antemortem hypoperfusion), and the VEGF level in the frontal cortex of the AD group was slightly elevated (indicating tissue hypoxia) [32].

Given the limited literature directly comparing the brain perfusion images of AD and VaD, we have combined insights from multiple sources. Although drawing a uniformly consistent conclusion is challenging, it is indeed possible to distinguish between the two disease groups based on specific data features. Recent research tends to view AD, VaD, and mixed dementia as different points on the same disease spectrum, suggesting that these conditions exhibit interconnected features. There remains a vast scope for deeper exploration in the future. Our study leverages both machine learning and deep learning techniques to distinguish between AD and VaD more sensitively using high-dimensional features, underscoring its feasibility. In the future, the goal is to predict and differentiate between various types of dementia with greater precision through combinations of these features.

## 5. Conclusions

This study proposed a method that can quickly use machine learning and deep learning models to automatically extract features based on a small amount of general clinical data and images to objectively distinguish AD and VaD. After filtering through SVM RFE with a 5-fold feature selection, the physiological data achieved an average accuracy of 81.22% in differentiating AD from VaD, with an AUC of 0.836. The image-based model, trained using the Inception V3 architecture, yielded an average accuracy of 85% and an AUC of 0.95. Additionally, Grad-CAM heatmaps were used to interpret the model’s decision-making, revealing significant differences between AD and VaD. Compared with the traditional analytical method, in the case of the AD group, both methods highlighted the precuneus and cuneus regions. In contrast, for the VaD group, there was no clear corresponding brain region. This approach underscores the potential of advanced analytical techniques in sensitively distinguishing between dementia types. In future work, we aim to combine these high-dimensional features to predict and differentiate various dementia subtypes more accurately, contributing to improved diagnostic precision in dementia care.

## Figures and Tables

**Figure 1 diagnostics-14-00365-f001:**
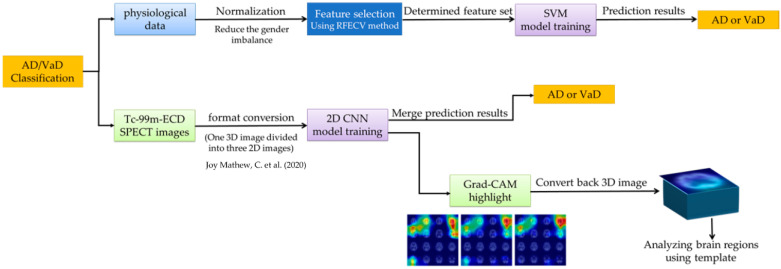
The flowchart of the entire data training process in this study.

**Figure 2 diagnostics-14-00365-f002:**
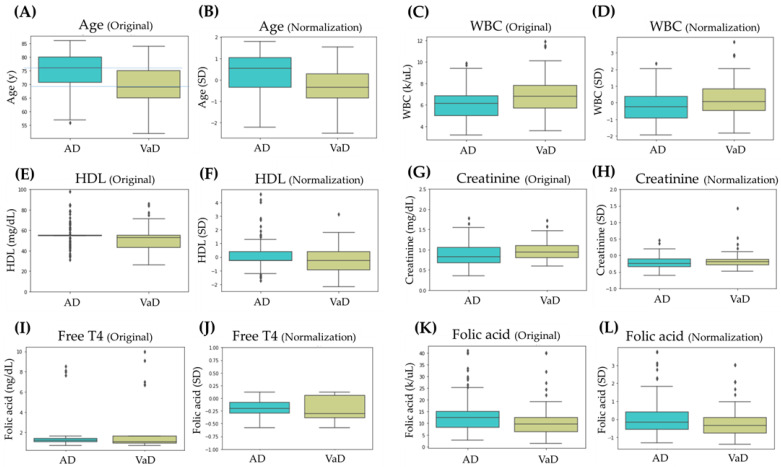
The boxplots compare six features before and after normalization. (**A**,**B**) are distributions of Age; (**C**,**D**) are distributions of WBC; (**E**,**F**) are distributions of HDL; (**G**,**H**) are distributions of creatinine; (**I**,**J**) are distributions of free T4; (**K**,**L**) are distributions of folic acid. Outliers are indicated by a “+” symbol.

**Figure 3 diagnostics-14-00365-f003:**
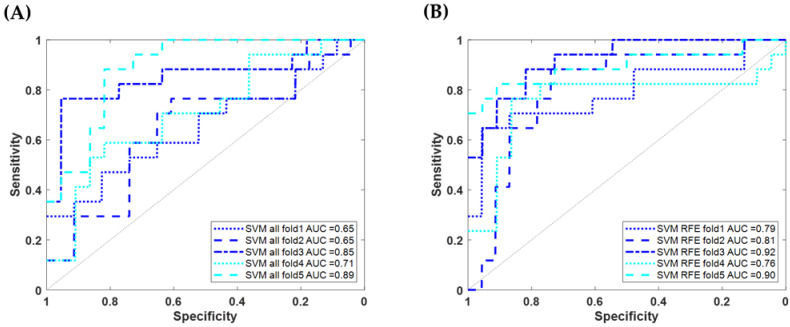
ROC curves of SVM model trained on physiological data for AD/VaD. (**A**) using complete data features (33); (**B**) using RFE-selected features (19).

**Figure 4 diagnostics-14-00365-f004:**
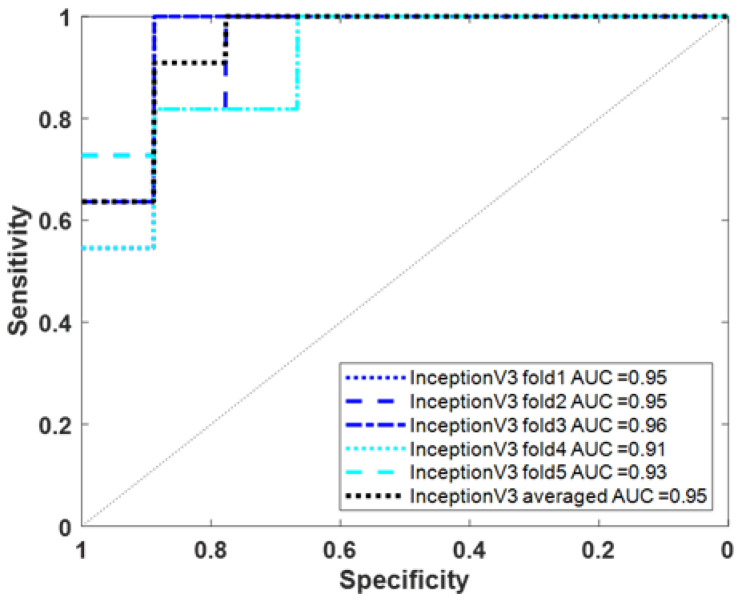
ROC curves of InceptionV3 model trained on ECD image for AD/VaD.

**Figure 5 diagnostics-14-00365-f005:**
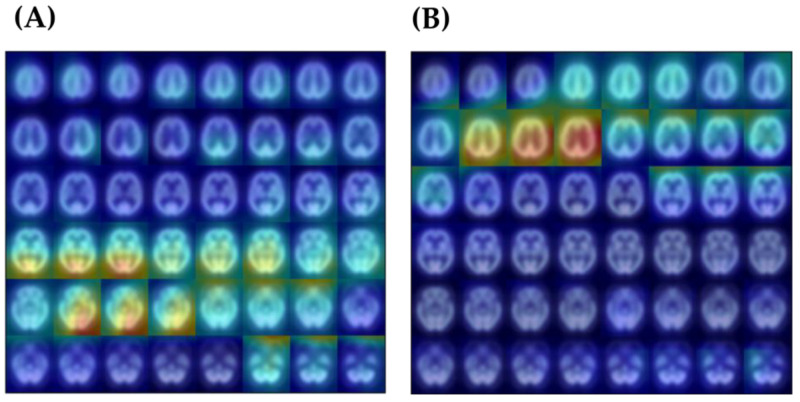
The average Grad-CAM heatmap for AD and VaD groups. (**A**) Average Grad-CAM map of AD testing set. (**B**) Average Grad-CAM map of VaD testing set.

**Figure 6 diagnostics-14-00365-f006:**
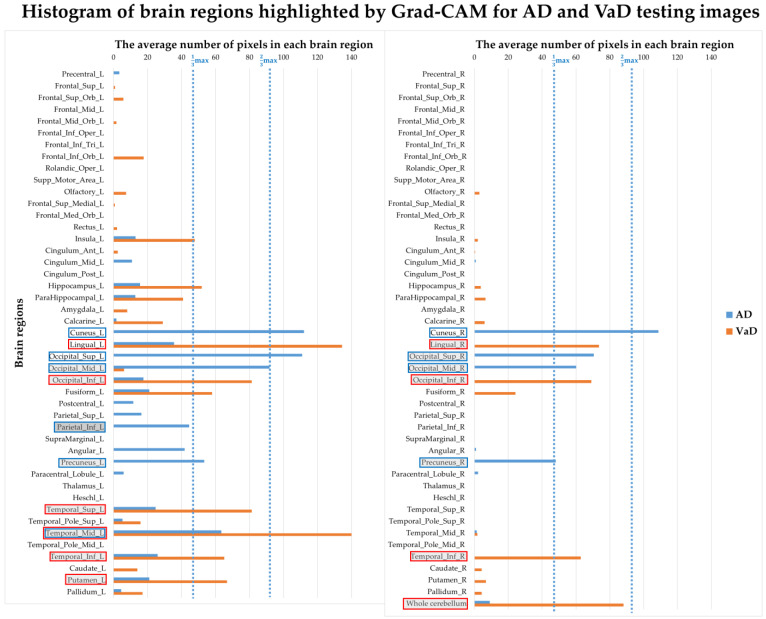
The histogram of pixel number in each brain region calculated from ‘lit-up’ areas from the Grad-CAM heatmap for AD and VaD groups. L, Left; R, Right; Sup, superior; Mid, middle; Inf, inferior; Med, medial; Ant, anterior; Post, posterior; Orb, orbital; Oper, operculum; Supp, supplementary; Tri, triangularis. The top ten are marked with square boxes, blue for the AD group and red for the VaD group. The square boxes with white background indicates greater than two-thirds of the maximum value; light gray represents greater than one-third of the maximum value; and dark gray represents less than one-third of the maximum value.

**Table 1 diagnostics-14-00365-t001:** The characteristics of demographic and physiological data.

	AD (*n* = 112)	VaD (*n* = 85)	*p*-Value (Ori)	*p*-Value (Norm)
**Basic information**				
Age (y), mean (SD)	74.5 (6.9)	69.7 (7.0)	**0.000 *****	**0.000 *****
Gender (Male), *n* (%)	53 (47.3)	57 (67.1)	**0.008**	**-**
Education (y), mean (SD)	8 (4.3)	8.7 (5.2)	**0.012**	**-**
FM with dementia, *n* (%)	27 (24.1)	5 (5.9)	**0.000 *****	**-**
FM with PD, *n* (%)	1 (0.9)	2 (2.4)	0.407	-
FM with stroke, *n* (%)	3 (2.7)	6 (7.1)	0.145	-
Exercise habits, *n* (%)	15 (13.4)	15 (17.6)	**0.012**	**-**
Sleep disorder, *n* (%)	27 (24.1)	10 (11.8)	**0.028**	**-**
Height (cm), mean (SD)	158.7 (7.5)	163 (8)	**0.000 *****	**0.041**
Body weight (kg), mean (SD)	59.2 (10.3)	63.8 (9.5)	**0.001**	**0.022**
Blood pressure (mmHg)				
Systolic, mean (SD)	130 (15.3)	133 (16)	0.184	0.152
Diastolic, mean (SD)	75.6 (9.9)	74.2 (9.6)	0.345	0.312
Heartbeat (bpm), mean (SD)	75.4 (11.7)	79.6 (12.2)	**0.016**	0.053
**Assessment scales**				
CDR (>0.5), *n* (%)	50 (44.6)	33 (38.8)	0.186	-
MMSE, mean (SD)	19.3 (5.2)	20 (6.6)	0.104	0.135
CASI, mean (SD)	64.7 (17.1)	67.8 (21.2)	**0.047**	0.075
**Laboratory examinations**				
Blood count (k/µL)				
Hb, mean (SD)	13.2 (1.5)	13.4 (1.8)	**0.016**	0.845
WBC, mean (SD)	6.1 (1.4)	6.9 (1.8)	**0.000 *****	**0.003**
Plt, mean (SD)	220.3 (71.1)	223.2 (61.1)	0.361	0.085
Glucose AC (mg/dL), mean (SD)	112.7 (31.2)	119.6 (37.8)	0.354	0.581
HbA1c (%), mean (SD)	6.2 (0.8)	6.4 (1.2)	0.967	0.872
Triglyceride (mg/dL), mean (SD)	121.3 (63.1)	140.7 (74.5)	0.074	0.057
Cholesterol (mg/dL)				
Total cholesterol, mean (SD)	184.4 (27.8)	179.2 (39.6)	0.151	0.471
HDL, mean (SD)	57.3 (15.1)	50.2 (11.5)	**0.000 *****	**0.007**
LDL, mean (SD)	114.3 (28.8)	106.2 (33)	0.148	0.287
Liver fuction index (U/L)				
GOT, mean (SD)	27.3 (10.4)	26.6 (18.3)	0.133	0.106
GPT, mean (SD)	24 (14.7)	24.5 (13.9)	0.693	0.929
BUN (mg/dL), mean (SD)	18.7 (9.7)	19.2 (12.5)	0.966	0.749
Creatinine (mg/dL), mean (SD)	1.1 (1.5)	1.2 (1.2)	**0.007**	**0.027**
TSH (mU/L), mean (SD)	3.1 (6.2)	2.5 (4.1)	0.276	0.709
Free T4 (ng/dL), mean (SD)	1.5 (1.3)	1.6 (1.6)	0.065	**0.036**
VitB12 (pg/mL), mean (SD)	959.8 (873.6)	742.5 (481.9)	0.500	0.861
Folic acid (ng/mL), mean (SD)	14.2 (8.5)	11.1 (7.1)	**0.002**	**0.014**

AD, Alzheimer’s Disease; VaD, vascular dementia; ori, original; norm, normalization; FM, family history; PD, Parkinson’s Disease; CDR, Clinical Dementia Rating; MMSE, Mini-Mental State Examination; CASI, Cognitive Abilities Screening Instrument; Hb, Hemoglobin; WBC, White Blood Cell; Plt, Platelet count; HbA1c, Hemoglobin A1c; HDL, High-Density Lipoprotein; LDL, Low-Density Lipoprotein; GOT, Glutamic Oxaloacetic Transaminase; GPT, Glutamic Pyruvic Transaminase; BUN, Blood Urea Nitrogen; TSH, Thyroid-Stimulating Hormone; VitB12, Vitamin B12. In the table, “***” indicates that the *p*-value is too small to display. Table cells with a background color and bold values denote columns and values where the *p*-value is significantly different.

**Table 2 diagnostics-14-00365-t002:** Comparison of the training performance using 33 and 19 features of physiological data for distinguishing between AD and VaD.

	Method		Sensitivity (%)	Specificity (%)	Precision (%)	Accuracy (%)	F1 Score (%)	AUC for AD/VaD (95% CI)
Physiological Data	SVM using 33 features	fold1	47.06 (8/17)	82.61 (19/23)	66.67 (8/12)	67.50 (27/40)	55.17	0.65 (0.47~0.81)
		fold2	29.41 (5/17)	82.61 (19/23)	55.56 (5/9)	60.00 (24/40)	38.46	0.65 (0.44~0.80)
		fold3	70.59 (12/17)	95.45 (21/22)	92.31 (12/13)	84.62 (33/39)	80.00	0.85 (0.62~0.95)
		fold4	58.82 (10/17)	68.18 (15/22)	58.82 (10/17)	64.10 (25/39)	58.82	0.71 (0.49~0.85)
		fold5	47.06 (8/17)	90.91 (20/22)	80.00 (8/10)	71.79 (28/39)	59.26	0.89 (0.76~0.97)
		**Averaged**	**50.59**	**83.93**	**70.49**	**69.54**	**58.90**	**0.75**
	SVM using 19 features	fold1	64.71 (11/17)	86.96 (20/23)	78.57 (11/14)	77.50 (31/40)	70.97	0.79 (0.59~0.92)
		fold2	64.71 (11/17)	86.96 (20/23)	78.57 (11/14)	77.50 (31/40)	70.97	0.81 (0.61~0.92)
		fold3	76.47 (13/17)	90.91 (20/22)	86.67 (13/15)	84.62 (33/39)	81.25	0.92 (0.80~0.97)
		fold4	76.47 (13/17)	81.82 (18/22)	76.47 (13/17)	79.49 (31/39)	76.47	0.76 (0.55~0.92)
		fold5	76.47 (13/17)	95.45 (21/22)	92.86 (13/14)	87.18 (34/39)	83.87	0.90 (0.69~0.97)
		**Averaged**	**71.76**	**88.39**	**82.43**	**81.22**	**76.73**	**0.836**

The 33 features are the items in Table 1. The 19 features are FM with dementia, FM with stroke, CDR, age, CASI, HDL, diastolic, total cholesterol, height, LDL, education, folic acid, VitB12, systolic, exercise habits, MMSE, WBC, heartbeat and creatinine. The average results of 5-fold are indicated in bold.

**Table 3 diagnostics-14-00365-t003:** Comparison of the training performance of ECD data sets in 5-fold for distinguishing between AD and VaD.

	Method		Sensitivity (%)	Specificity (%)	Precision (%)	Accuracy (%)	F1 Score (%)	AUC for AD/VaD (95% CI)
ECD Image	InceptionV3	fold1	63.64 (7/11)	88.89 (8/9)	87.50 (7/8)	75.00 (15/20)	73.68	0.95 (0.69~1.00)
		fold2	90.91 (10/11)	77.78 (7/9)	83.33 (10/12)	85.00 (17/20)	86.96	0.95 (0.77~1.00)
		fold3	100.00 (11/11)	88.89 (8/9)	91.67 (11/12)	95.00 (19/20)	95.65	0.96 (0.77~1.00)
		fold4	90.91 (10/11)	66.67 (6/9)	76.92 (10/13)	80.00 (16/20)	83.33	0.91 (0.69~0.98)
		fold5	72.73 (8/11)	88.89 (8/9)	88.89 (8/9)	80.00 (16/20)	80.00	0.93 (0.69~1.00)
		**Averaged**	**81.82**	**88.89**	**90.00**	**85.00**	**85.71**	**0.95 (0.73~1.00)**

**Table 4 diagnostics-14-00365-t004:** The top ten brain regions from Grad-CAM heatmap and SPM analysis from AD and VaD groups.

	Grad-CAM Map of Inception V3 Model		Image Analysis by SPM	
Brain Region Importance Ranking	AD	VaD	AD	VaD
1	**Cuneus_L**	Temporal_Mid_L	Thalamus_L	Paracentral_Lobule_L
2	Occipital_Sup_L	Lingual_L	Thalamus_R	Postcentral_R
3	**Cuneus_R**	Whole cerebellum	Lingual_R	Precentral_L
4	Occipital_Mid_L	Temporal_Sup_L	Calcarine_R	Supp_Motor_Area_R
5	Occipital_Sup_R	Occipital_Inf_L	Hippocampus_R	Precentral_R
6	Temporal_Mid_L	Lingual_R	**Precuneus_R**	Frontal_Sup_R
7	Occipital_Mid_R	Occipital_Inf_R	ParaHippocampal_L	Supp_Motor_Area_L
8	**Precuneus_L**	Putamen_L	ParaHippocampal_R	Parietal_Sup_R
9	**Precuneus_R**	Temporal_Inf_L	Hippocampus_L	Paracentral_Lobule_R
10	Parietal_Inf_L	Temporal_Inf_R	**Cuneus_R**	Frontal_Sup_L

L, Left; R, Right; Sup, superior; Mid, middle; Inf, inferior. The bolded text indicates repeated items in the brain regions. The color coding is a ranking of the number of pixels in each brain region calculated from ‘lit-up’ areas from the Grad-CAM heatmap for AD and VaD groups: a white background indicates greater than two-thirds of the maximum value; light gray represents greater than one-third of the maximum value; and dark gray represents less than one-third of the maximum value.

## Data Availability

The data presented in this study are available on request from the corresponding author.

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
