# Peer review of "Classification Prediction of Alzheimer’s Disease and Vascular Dementia Using Physiological Data and ECD SPECT Images"

_diagnostics, 2024, doi:10.3390/diagnostics14040365_

Round 1

Reviewer 1 Report

Comments and Suggestions for Authors

The article is well written and the research is useful for the society.

The discussion section is well written but the conclusion is too short. Elaborate the conclusion with the outcome of the research and include major future scopes.

Figure 1: After feature selection, directly the prediction result is mentioned. Check.

Explain RFECV method of feature selection in short and tell the need/significance of the same.

Reviewer 2 Report

Comments and Suggestions for Authors

Thank you for giving me the opportunity to review the manuscript entitled "Classification Prediction of Alzheimer’s disease and Vascular Dementia using Physiological Data and ECD SPECT Images" in which the authors present a SVM-based algorithm to differentiate between AD und VaD. To Train and test the model, the Taiwanese Nuclear Brain Imaging Database.

The manuscript is well written and structured, material and methods are described clearly and the results are presented in a structured way. The figures and tables are of good quality.

However, there are some concerns that need to be addressed:

- in the introduction please mention the benefit and the impact of arterial spin labeling for the differential diagnosis of AD 

-p3l114 - M.-C.P - please explain in more details; also add references for the database, that demonstrate the representative values of the database you mentioned in the manuscript. How many datasets are included in the database? What percent of the patient in the database was used for this study or was the entire database used? If not, how did you select the patients.

- although it is a common disease, the number of subjects for training and testing is relatively small. Why did the authors not include a larger number of datasets?

- IRB - the approval was for the database or for the presented study? Was informed consent necessary?

-p3l134 - 57:28 - numbers or percent?

Comments on the Quality of English Language

no editing necessary
